# Analysis of Influence of Segmentation, Features, and Classification in sEMG Processing: A Case Study of Recognition of Brazilian Sign Language Alphabet

**DOI:** 10.3390/s20164359

**Published:** 2020-08-05

**Authors:** José Jair Alves Mendes Junior, Melissa La Banca Freitas, Daniel Prado Campos, Felipe Adalberto Farinelli, Sergio Luiz Stevan, Sérgio Francisco Pichorim

**Affiliations:** 1Graduate Program in Electrical Engineering and Industrial Informatics (CPGEI), Federal University of Technology–Paraná (UTFPR), Curitiba (PR) 80230-901, Brazil; josej@alunos.utfpr.edu.br (J.J.A.M.J.); felipe.farinelli@outlook.com (F.A.F.); pichorim@utfpr.edu.br (S.F.P.); 2Graduate Program in Electrical Engineering (PPGEE), Federal University of Technology–Paraná (UTFPR), Ponta Grossa (PR) 84017-220, Brazil; melissa.1995@alunos.utfpr.edu.br; 3Graduate Program in Biomedical Engineering (PPGEB), Federal University of Technology–Paraná (UTFPR), Curitiba (PR) 80230-901, Brazil; danielcampos@utfpr.edu.br

**Keywords:** sign language, surface electromyography, wearable device, signal segmentation, feature extraction, pattern recognition, machine learning

## Abstract

Sign Language recognition systems aid communication among deaf people, hearing impaired people, and speakers. One of the types of signals that has seen increased studies and that can be used as input for these systems is surface electromyography (sEMG). This work presents the recognition of a set of alphabet gestures from Brazilian Sign Language (Libras) using sEMG acquired from an armband. Only sEMG signals were used as input. Signals from 12 subjects were acquired using a Myo^TM^ armband for the 26 signs of the Libras alphabet. Additionally, as the sEMG has several signal processing parameters, the influence of segmentation, feature extraction, and classification was considered at each step of the pattern recognition. In segmentation, window length and the presence of four levels of overlap rates were analyzed, as well as the contribution of each feature, the literature feature sets, and new feature sets proposed for different classifiers. We found that the overlap rate had a high influence on this task. Accuracies in the order of 99% were achieved for the following factors: segments of 1.75 s with a 12.5% overlap rate; the proposed set of four features; and random forest (RF) classifiers.

## 1. Introduction

Sign Language recognition (SLR) systems have an important task in aiding the communication between deaf people and the hearing impaired. These systems recognize Sign Language, a structured set of manual gestures that are used by the deaf and hearing impaired population as a communication system [1]. It is known that social communication is one of the barriers that people with hearing disabilities face, especially when they need to establish communication with people that do not know Sign Language (SL) [2]. In many cases, even if sign language is an official (native) language that allows the cognitive development of an individual [3], the communication between deaf and speaking people presents some failures when the speakers do not have enough knowledge of SL [4], which represents the majority of the speakers. Thus, an SLR system translates the SL signs to text or speech to ease the communication between people with deafness and the speakers [5,6].

There are two main technologies of SLRs: ones based on images (which use cameras to capture the signals), and ones based on sensors (which use sensors placed on the upper limbs or wearable devices, such as wristbands, gloves, and armbands) [1,7]. Although the techniques with an image approach have been widely used, they are highly dependent on environmental factors, such as fixed placement of the camera, brightness and contrast. On the other hand, sensor-based SLR systems use motion devices, inertial sensors, and electrical biosignals, such as surface electromyography (sEMG), as input. These devices are a relevant proposal because they have more robustness since the signal can be acquired in any environment. In particular, sEMG signals have seen recent advances due to the development of wearable devices [8] and data processing techniques [9] for gesture recognition [10]. Wearable sEMG sensors-based SLR/gesture recognition systems have become more popular in recent years because of they are low cost, privacy nonintrusive, and have a ubiquitous sensing ability compared to vision-based approaches [11]. Nevertheless, it is necessary to mention that this technique has some limitations for SLR systems. Wearing a device can limit the action of a subject during the performance of a sign (especially if the device causes some discomfort for the subject), which restricts some applications for laboratory or controlled environments [1], and sEMG can generate data with a noisy aspect [12], which are difficult to be visually interpreted. Furthermore, in SLR applications, some signs combine hand movements and facial expressions, requiring the use of complementary devices to both acquire facial expressions and perform signal recognition.

sEMG is a technique used to acquire, process, and record electrical activity from skeletal muscles [13]. This technique uses surface electrodes placed on the skin. The obtained signal is a spatial-temporal sum of the action potentials that are generated in the muscles during their contraction, which is commanded by the nervous system [14]. The sEMG is useful for SLR, which is considered a gesture recognition application, because in contrast to other kinds of sensors (such as inertial and flexible resistors), the extracted signal presents data related to movement execution and its respective intention: one of the motives for its use is in applications for amputees [12]. Multichannel sEMG signals, recorded by a set of sEMG sensors placed on the arms, contain rich muscle information associated with different sign gestures [15].

During the processing of sEMG signals, segmentation, feature extraction, and classification stages are a key factor to guarantee high hit rates in pattern recognition [16]. However, these stages have several parameters that need great attention. In the segmentation, these include the muscle activation method, the window size, and the overlapping influence in the classification process [17]. The choice of feature sets to be extracted is the target for various scientific studies. New combination sets have been presented in the literature, seeking to improve accuracy and robustness for sEMG gesture recognition [18,19,20]. Several classification techniques are used, especially based on supervised machine learning algorithms [1], such as support vector machines (SVM) and artificial neural networks (ANN) [17,21]. For different applications, different features and different classifiers may exhibit significantly different performance. It is advisable to try different approaches to determine the best one.

Thus, this work presents the pattern recognition process used for the development of an SLR for Brazilian Sign Language (Libras) alphabet using sEMG signals as input. Libras was chosen due to it having similar gestures to other SLs and, in Brazil, this SL is recognized as an official language. Even though sEMG was explored for gesture recognition in other SLs around the world, such as American [11,22], Chinese [23,24], and Greek [10], there are few studies that apply for Libras. Regarding the sEMG signals, the influence of each processing stage is explored: the segmentation and windowing of sEMG signals, feature extraction, and their relation with the classification. In this study, the armband acquisition approach (using the commercial Myo^TM^ armband) was considered due to the recent advances in sEMG pattern recognition using these devices and their popularization in similar researches. These recognition systems enable real-time operation and, at the same time, are relatively simple to implement.

The hypothesis that sEMG features extracted from sliding window segmentation could recognize Brazilian SL is evaluated in this work. At the same time, the parameters of a sliding window are analyzed to achieve the best classification performance. Additionally, as most previous works considered the fusion of IMU (Inertial Measurement Unit) and sEMG, the efficiency of sEMG signals alone remains unclear. In this way, this work also brings some results of this modality of sensor to Libras recognition. Besides that, the effects of some parameters in the recognition process are little explored. For example, what is the influence of segmentation parameters, such as the window length and the overlapped segments, in an SLR system? Alternatively, which features (or feature sets) could be used to improve a sEMG-based SLR systems? Moreover, which classifier (or classifiers) could be more suitable for an sEMG armband device for Libras recognition? These questions arise in the process of SLR recognition, and in some related works these questions are not fully answered [11,23,25,26,27]. Thus, a methodology is presented in this paper to evaluate how the sEMG processing parameters influence classification systems for Brazilian Sign Language.

This work is organized as follows: Section 2 presents the related works evolving sEMG for SLR; Section 3 presents the data acquisition device, the protocol to record the sEMG signals, and the methodology to process the signals; Section 4 presents the results obtained for each methodology stage; and Section 5 presents the main conclusions of this work.

## 2. Related Works

In the gesture recognition field, the sEMG signal is widely explored and applied as an input [1]. In recent years, the number of works that apply sEMG as input for SLRs significantly increased. The majority of these works focused on American (ASL) and Chinese (CSL) Sign Languages and used multi-channel acquisition (more than one sEMG channel). Moreover, it is common to employ inertial sensors for data fusion to ease the sEMG segmentation and to increase the accuracy of classification in dynamic gestures.

The first studies for SLR and sEMG were conducted to classify nine signals from ASL using 16 features and discriminant analysis (DA) [25]. Two channels were placed on the forearm muscles, allowing hit rates of about 97%. However, there no information was presented about the number of subjects and their characteristics. Afterward, 40 [28] and 80 [11] signals from ASL were recognized, with a high number of features and classifiers. Four channels (fixed on the forearm muscles) and inertial sensors fusion were able to classify the gestures for four subjects with a 96% hit rate with several sEMG features.

Many works aimed for the recognition of signals from CSL. Zhang et al. (2011) [24] applied hidden Markov models (HMMs) to classify 72 signals using five channels and accelerometers in two subjects. In 2015, 223 gestures acquired from five subjects were classified with accuracies from 92% to 96% using HMM models and the dynamic time warping (DTW) technique [29]. Random forests (RF) and HMM were employed to recognize 121 phonemes from CSL for five subjects using five channels and an accelerometer sensor [27]. Moreover, Zhuang et al. (2017) [23] demonstrated that the window size interferes with the classification process. At least 18 signals acquired from eight subjects were classified using linear discriminant analysis (LDA), reaching a 91% accuracy level [23].

Concerning other SLs, 60 gestures from Greek Sign Language (GSL) were classified using discriminant analysis and sEMG sample entropy with 93% hit rates [10]. In this study, five channels were placed on the forearm of three subjects and data from the accelerometer were used for classification. Moreover, 30 gestures in Korean Sign Language (KSL) were classified for six subjects using the commercial Myo^TM^ armband [26]. The raw signals from the inertial sensors were inserted into a convolutional neural network (CNN), achieving a 98% accuracy level. Other works presented the recognition for Thai Sign Language (TSL) [30] and Indonesian Sign Language (ISL) [31,32]; however, in their methodologies, some aspects were not explained and described, such as the number of subjects from which the data were collected.

Regarding Libras, it is necessary to point out that image-approach was the tool for several studies for SLRs [33,34,35]. The sensor approach is based on instrumented gloves, and sEMG wearable devices are a field that is being explored in recent works, achieving high accuracies and hit rates [36,37]. There is a limited number of works combining sEMG and Libras, and their focus is on alphabet classification. Abreu et al. (2016) [38] used the Myo^TM^ armband to classify 20 static letters with an SVM classifier. The accuracy for each letter was presented instead of the overall accuracy; however, the average of hit rates in the test stage was about 41%. This work demonstrated that it is suitable to classify Libras from sEMG signals, but the authors pointed out that limitations in processing and acquisition hindered achieving a high performance. Posteriorly, the same process was repeated for 20 static letters by Mendes Junior et al. (2019) [39]. From one subject, seven time-domain and five frequency-domain features were extracted from the sEMG signals, and a multi-layer perceptron (MLP) neural network was used as the classifier. In this approach, the overall accuracy was above 80% during the test step. In this work, it the errors that can occur during the classification of these gestures were verified, such as the gesture “a” misclassified as “s”, “n” as “p”, and “g” as “w”. In sEMG applications, the evaluation of the influence of acquisition and processing steps, such as number of channels, segmentation methods, features extraction, and type of classifiers, is missing.

Some aspects can be raised concerning the characteristics of these works. The first is that this application can reach high hit rates; however, the number of subjects used in these studies was low (less than 10 subjects), which can explain the value of the accuracies. In data acquisition, the armband approach was been considered (e.g., Myo^TM^ armband) instead of the fixed electrode placement. In segmentation, the overlapped segmentation was applied in some works, but its variation was not evaluated. Even knowing that this parameter (together with window length) affects the pattern recognition process [17,40], the effect of overlapping the segments on accuracy was not explored and was usually based on a fixed length of overlapped segments. During the classification, the contribution for each feature or the use of specific feature sets and their influence on accuracy were not evaluated. The most repeated features were mean absolute value (MAV) and autoregression (AR), but this does not explain why these features were used rather than others available in sEMG processing. Regarding the number of classifiers, different techniques were used, but the behavior for more than four techniques was not presented.

Other recent studies presented frameworks and techniques that have been developed to solve the SLR problem, focusing on signal processing for sensors with different natures. For example, a leap motion sensor was used with a hidden Markov layer for pattern recognition of 24 gestures of ASL [41] and instrumented gloves with flex [42] and/or inertial sensors [43] with machine learning techniques reached accuracies ranging from 86% to 98%. A combination of images and deep learning techniques demonstrated high performance for this task, such as for Persian Sign Language [44] and CSL [45]. This indicates that this topic has been developed recently. The most prominent similarity shared by studies with the sEMG studies was signal processing, which is a recurring demand for this application.

## 3. Materials and Methods

In this section, the data acquisition device, the experimental protocol, the processing technique, and the classification methods are presented.

### 3.1. Data Acquisition

The Ethical Committee for Research in Humans of the Federal University of Technology—Paraná (CAAE 89638918.0.0000.5547) approved the data acquisition. The chosen gesture set was the Libras alphabet, composed of 26 gestures (20 static gestures and six dynamic gestures) presented in Figure 1. It is a closed set of gestures, which is used by Libras speakers to spell words, acronyms, names, and facilitate the construction of other signals (e.g., the signal related to the name of a person or technical or specialized vocabulary) [34]. In addition, these signals use different hand configurations and some movements, which are suitable for gesture recognition applications.

The commercial Myo^TM^ armband was chosen for data acquisition, and is presented in Figure 2a. This device is widely used in gesture recognition research [46,47]. Myo^TM^ has eight equidistant channels, with 200 samples/s, 8-bit resolution from ADC (Analog to Digital Converter), and wireless communication via Bluetooth [48]. Signal acquisition is made through dry stainless-steel electrodes and there is a 3-axis sensor unit with an accelerometer, gyroscope, and magnetometer. Myo^TM^ was placed on the right forearm of subjects (Figure 2b), which is the orientation needed to place channel 3 on the *flexor carpi ulnaris* muscle. All the subjects were right-handed and the gestures were performed only with the right hand.

Signals from 12 subjects (10 men and two women), with an average age of 23.2 ± 2.4 years, were acquired (for more information, see Appendix A). The subjects were healthy, without myopathies or muscle diseases in the upper limbs. Figure 3 presents the signal acquisition process. A graphical interface was developed to inform subjects what gesture should be performed and the next gesture in the sequence. When the subject was required to perform a gesture, a light indicator turned on. The indicator turned off when the subject was required to rest their forearm muscles. In each change, a buzzer sounded. The time interval for each gesture was defined with a frequency of 45 beats per minute (bpm), equivalent to 1.3 s between performing each gesture and rest state. In each trial, the subjects performed the 26 gestures; this sequence was repeated 20 times for each subject.

### 3.2. Signal Processing

The sEMG processing for gesture classification can be divided into three main stages: segmentation, feature extraction, and pattern recognition.

#### 3.2.1. Segmentation

Segmentation is responsible for detecting and windowing the sEMG signal. In this work, the detection of the gesture was made using the double threshold onset technique. The onset was based on the energy for the eight channels, calculated as described in [27]. The threshold was defined as 10% of the maximum energy for each trial compared with a time interval of 150 ms to avoid false triggers and noises [49].

In segmentation, two main parameters were evaluated: the window length and the overlap rate. The window range was defined from 0.25 to 2.25 s, with steps of 0.25 s. These values were defined taking into account the delay for sEMG processing [50,51] and the time at which the gestures were performed in the acquisition. To analyze the overlapping, the parameter overlap fraction (*ϕ*) was calculated by:
*ϕ* = *k*/*W*,
(1)
where *W* is the window length of each segment and *k* is the time step from one segment to the next. This means that an overlap fraction of 1 (100%) represents a disruptive window, 0.5 (50%) a half-step overlapping window, 0.25 (25%) a quarter-step overlapping window, and 0.125 (12.5%) an eighth-step overlapping window. Four parameters were analyzed during overlapping: 100% (a disruptive window), 50%, 25%, and 12.5%. This method is more effective than the use of a fixed step for overlap signals for evaluation due to having different window segments to analyze. The overlapping process may also include information that is present in the adjacent signal window; therefore, it works as a redundant extraction to ensure that the useful information is contained inside the analysis segment [51].

#### 3.2.2. Feature Extraction

Feature extraction is the stage where the dimensionality from the segments is reduced into useful information for classifiers, especially for the sEMG signals that have a stochastic nature [52]. The features analyzed in this work are presented in Table 1 with their respective parameters for implementation. In total, 33 features were extracted and evaluated: 24 features from the time domain and nine from the frequency domain. These features were chosen due to their use in other feature sets in the literature [19,53,54,55], which present their math expressions. Frequency features were extracted after the signals passed to a fast Fourier transform in each segment with 2000 points of resolution. Some features needed parameterization and their values were based on studies that applied them for gesture recognition. For auto-regressive and cepstral coefficients, the fourth order was chosen [53,56]; in the histogram feature, nine bins were extracted [57]. The second-moment was used in L-Scale [55]; the third order was chosen for V-order [53]; and a threshold value of 10^−2^ was defined for the Myopulse percentage rate, Willison amplitude, sign slope change, and zero crossing features [58]. A 0.2 σ tolerance (standard deviation) and two dimensions were chosen for sample entropy [59]; the band of frequency was split at the average for the frequency ratio [53].

The performance of the features in the classification process was also compared with feature sets presented in the literature [54,55,56]. The features in Table 1 were analyzed individually and organized in feature sets based on their accuracy in the pattern recognition process. Some groups were tested: a set with the most repeated features found in related works (Section 2), a set of the best features with high performance individually, a set with the best time-domain features, a set with the best frequency-domain features, and a reduced set of features with relevance to accuracy and statistical analysis. The features sets are summarized in Table 2.

#### 3.2.3. Classification

In the pattern recognition stage, classifiers label the gestures based on the inputs. In this work, the following classifiers were applied: *k*-nearest neighbor (KNN), linear discriminant analysis (LDA), probabilistic naïve Bayes (NB), quadratic discriminant analysis (QDA), extreme learning machine (ELM) and multi-layer perceptron (MLP) neural networks, random forest (RF), and support vector machines based on linear discrimination (SVMLin) and Gaussian kernel-type based on radial basis function (SVMRBF). These classifiers were selected based on the classifiers listed in related works (Section 2) and presented high accuracies in gesture classification [27,53,60,61]. The *k*-fold cross-validation method was used, using 10 folds.

Regarding the classifiers, some of them need parameterization, which was chosen using the cross-validation procedure. One nearest neighbor was used in KNN. The hyperbolical function was used in the neural networks and the number of neurons in the hidden layer for the ELM was 1000 and in MLP 30. In random forests, 30 trees were selected. The SVMs were built based on the LibSVM library [62]. In SVMLin, the *C* parameter was defined as 100; in SVMRBF *C* was programmed as 10 and the size of Gaussian functions was defined as 1. Table 3 summarizes their parameterization.

### 3.3. Methodology to Data Processing

A methodology was developed to decrease the computational effort to find the parameters of this SLR. Figure 4 presents the steps performed for data processing. Firstly, the data were segmented in a fixed window length and an overlap rate with 100%, which means that the segments are windows without overlap (disruptive segments). These values in segmentation, and especially the window length and no presence of overlapping, were chosen due to high accuracies in similar works. Thus, the analysis was performed in the features extracted in Table 1.

The features were classified individually and the best features and the classifiers with high accuracies were selected. These features compounded the aforementioned groups from G6 to G8.

The second stage was the analysis for the feature sets (G1 to G8), particularly the features presented in the literature (G1 to G5) and the features in the previous stage (G6 to G8). The best feature sets were selected for segmentation analysis, which was the third stage. The accuracies for ranges of window length and overlap rate values were calculated for the selected classifiers, searching for the best combination of these two parameters. As aforementioned in Section 3.2.3, the pattern recognition was performed using k-fold cross-validation (with k equals to 10) for the entire dataset in the analysis from stages 1 to 3. Thus, this method was chosen to verify the influence of the parameters for the entire dataset, without taking into account the difference between the subjects.

The last stage was the analysis of accuracy for each subject. Two methods were used to evaluate each subject. Firstly, the data from each subject were evaluated separately; in each training and test samples were from the same subject. Afterwards, the training and test sets were separated based on the acquisition trials, with k-fold cross-validation based on the acquisition trials (Figure 3). For example, *k* number of trials were removed for training data for each subject. The test set was composed of these *k*-trials samples and tested individually for each subject. This analysis can show how the pattern recognition system behaves for samples from the subjects with a previous calibration (provided for the training samples) while also considering all the full datasets and involving all subjects. As it is based on a calibration that occurs in a wearable device for sEMG (the Myo^TM^) [63], it provides a generalist vision for the pattern recognition process. In both analyses, the data were partitioned using a 10-fold cross validation method.

## 4. Results and Discussion

The results are presented following the stages described in the methodology (Figure 4). Firstly, the accuracy for each feature and classifier are presented individually, followed by the analysis of feature sets. Posteriorly, the influence of segmentation is considered and the tests for each subject are exhibited. Finally, the results are compared with those presented in the literature (related works).

### 4.1. Feature and Classifier Separately

The first step consisted in the evaluation of the behavior of each feature individually. Figure 5 presents the accuracies for the features (presented in Table 1) extracted individually for all the classifiers. In this analysis, the window length and overlap rate were set as 1 s and 100% (a disruptive window without overlap), respectively. It can be noted that there are features that have high accuracies comparing to others, which is common in sEMG processing. Moreover, the use of an isolated feature cannot provide an acceptable level of accuracy. Some features, such as in the frequency domain (e.g., Mean Frequency, Median Frequency, and Frequency Ratio) and with complex information (e.g., 4-th order Autoregressive Coefficients and Sample Entropy) had accuracies close to prior probability (1/26), which are not being indicated in this application. Features with a high correlation of amplitude levels, such as in the time domain, which presented performances of up to 50% for some classifiers, were indicated more in this case. This reinforces a feature set needs to be built.

A mean for the performance of all classifiers was extracted (presented by the dashed lines) to identify the best features and classifiers in this scenario. Regarding the classifiers, one can note that four of them had accuracies of up to the mean: KNN, ELM, RF, and SVMRBF. This behavior can be noted in similar works for sEMG signals that were compared with different classifiers [53,64]. Because of these values, these four classifiers were analyzed more comprehensively, aiming to reduce the computational effort.

In decreasing order of hit rates, the features with better performance were MFL, MNP, TTP, RMS, SM1, LS, DASDV, SM2, SM3, IEMG, MAV, WL, MSR, MAV1, HIST, VAREMG, and MAV2. Based on these features, the feature sets G6 to G9 were organized as previously mentioned. G6 was composed of all 17 features. G7 was composed of the best 12 features in the time domain: MFL, RMS, LS, DASDV, IEMG, MAV, WL, MSR, MAV1, HIST, VAREMG, and MAV2. G8 was composed of MNP, SM1, SM2, SM3, and TTP.

The feature set G9 was composed of the four best features from G6 that demonstrated high classification accuracy with a reduced number of features. The reduction was based on the statistical dependency of the distribution after the pattern recognition process (using a Friedman test and Tukey’s post-hoc test for multi-comparison, *p* > 0.05) and the following features were selected: MFL, MNP, TTP, and RMS. This process is presented in Figure 6, composed of the features with the highest accuracies in G6 and the best four classifiers (ELM, KNN, RF, and SVMRBF). Figure 6a presents the accuracy rates obtained with the increase of each feature for the classification. It can be noted that when increasing the number of features, their medians stabilized for four features (up to Root Mean Square, RMS). After this, there was no significant gain and the coverage of distribution was more concentrated. In Figure 6b, it can be noted that the distribution of these four features overlaps for almost all distributions (except the first). The four features presented equivalent distribution when more features were inserted into the classification (Figure 6b), besides that the range of their distribution (denoted by the whiskers of Figure 6a) covered the values when more features were introduced into the classification. With these definitions, the next analysis was the influence of the feature sets.

### 4.2. Analysis of Feature Sets

Figure 7 presents the classification performance for the feature sets from the literature (G1 to G5) and the features with high hit rates in the previous section (G6–G9). The KNN, ELM, RF, and SVMRBF classifiers were considered in this analysis due to their high accuracies in the individual feature analysis. It can be noted that the accuracies show a slight increase using feature sets compared to the use of individual features, with means ranging from 40% to 60%. The best classifier for the feature sets alternated between RF and SVMRBF.

The classical feature sets (G1 and G2) and the most repeated features in related works (G3) presented the worst performance compared with TD4 (G4), TD9 (G5), and the features established in the previous section. This could be explained due to the presence of features that cannot have high separability for the sampling rate used in the Myo^TM^ device (such as ZC and SSC) [55] or have complex features (such as AR4).

Even though the best accuracies were present from G4 to G9, it was necessary to verify that they belonged to the same distribution. The accuracy distributions were submitted to a Tukey post-hoc from Friedman test, as presented in Figure 7b. The ranks presented the distribution, and the same groups were analyzed with a confidence interval of 95% (*p*-value < 0.05). There were two main distributions. G1, G2, and G3, with the lowest accuracies, belonged to the same distribution. From the feature sets with the highest accuracies, G4, G5, G6, G8, and G9 belonged to the same distribution, which was statistically different from the other feature sets.

Comparing the feature sets, three of them were considered for the subsequent analysis: G4 (TD4), G8 (best in the frequency domain), and G9 (reduced set from the best features). G5 was discarded because it had three of the four features of G4, which had a similar distribution. Considering the number of features, G5 had nine features whilst G4 had only four. This represents a decrease in the time used to extract these features, considering an embedded application. The same criterion was used to discard group G6, which had 17 features whilst G8 and G9 had five and four features, respectively.

### 4.3. Influence of Segmentation Parameters

As the previous analysis consisted of fixed values for window length and overlap rate, the next step, shown in Figure 4, was the variation of segmentation parameters. Figure 8 presents the results obtained for the variation of the overlap and the window size for the classifiers (ELM, KNN, RF, and SVMRBF) and the feature sets with high performance (G4, G8, and G9). Each color represents a level of accuracy, as demonstrated by the color bar. Dark blue represents the lowest accuracy whilst light yellow represents the highest one.

The first fact to note is that an increase in the number of elements in the segment also increased the hit rates. This fact occurred for all classifiers, feature sets, and overlap rates. As mentioned before, in works related to sEMG pattern recognition, the window length directly affected the accuracies [17,65,66]. However, the window size cannot be too extensive or large and thus compromise the whole application (involving the detection of sEMG and the final response of the actuation). The accuracies showed significant increases for segments longer than 1 s (above 200 datapoints in each segment). The inverse occurred for overlap rate: no overlap (100%) or a high rate resulted in low accuracies, from 60% to 70% for both classifiers and feature sets. It is notable how the overlap process influenced the accuracy in every scenario. Then, it is necessary to highlight that for window lengths in the order of 0.25 and 0.5 s, some combinations, such as G4 for KNN and RF, presented hit hates in the order of 80 to 90% for 25% and 12.5% overlaps. Even though this was the highest accuracy that the recognition process could reach, it is an expressive value.

ELM and the G8 feature set had the lowest accuracies. The other classifiers had similar results, both for overlap and window length, and G4 and G9 presented similar distributions. Still, a window length of 2.25 s had high accuracy; such a length needs to search for a window length with the same distribution with a lower number of segments in a window. An analysis in groups was performed, using a Friedman test and a Nemenyi post-hoc test (*p* > 0.05) and considering the groups containing the segmentation parameters. Based on their results, critical distance diagrams (CD) were performed and are presented in Figure 9. These diagrams list the groups with the best ranks on the left and distributions without statistical difference are connected with a solid line [67]. Figure 9a presents the comparison related to the classifiers and window length. It can be noted that segments of 1.75 s have similar distributions to 2.25 s and 2 s, even though the latter two present the best performance rates. Figure 9b presents the analysis of classifiers and feature sets from the overlap rate. The overlap rate of 12.5% showed ranks placed differently than 25%; however, there were areas of overlap between them.

Due to overlapping found in the groups of Figure 9, the distributions were analyzed separately. Table 4 presents the similar accuracies in the segmentation analysis for the observed similar parameters, such as window length for 1.25 to 2.25 s and overlap rate for 25% and 12.5%. It should be highlighted that using 2.25 s, the classifiers had 99% accuracy; for some classifiers, such as SVMRBF and KNN, one second less only decreased the accuracy by 1% for a 12.5% overlap rate.

Aiming to verify if they belong to the same distributions, window length data (each column) were submitted to a Tukey post-hoc from Friedman test and the overlap data (row) were submitted to a Wilcoxon test. These analyses are also presented in Table 4 with the following indices: a (1.25 s), b (1.5 s), c (1.75 s), d (2 s), and e (2.25 s) are window length and * indicates the same distribution for overlap rate. For example, when a value is marked with (a), it means that its distribution is statistically significant for 1.25 s. The same process was performed for overlap rate analysis: when the value is marked with (*), it means that 25 and 12.5 have the same distribution.

Searching a segment with distributions equivalent to the obtained results for 2.25 s with a lower number of segments, the similar distribution found was for 1.75 s. Regarding the overlap rate, one can note that there are still similar values of accuracies; the distributions presented statistical differences for the interval of 5% in the Wilcoxon test. Separately, the 25% and 12.5% overlapping ratios had different distributions in several cases, which means that these parameters can be investigated in the individual subject analysis.

### 4.4. Analysis of Subjects

After the analysis of the subjects, it is necessary to attempt using the feature sets. One can note that G4 and G9 did not present significant differences between them, which was statistically verified. To choose one feature set for the subject analysis, the computation time was measured for both feature sets. Even though G9 has frequency-domain features (which means calculating a frequency transform), it is 20% faster than G4. It should be explained due to the features present in G4, it has more calculation complexity than G9 (e.g., as LS and MFL) that can represent a computational cost in embedded applications.

Considering G9 and the aforementioned segmentation parameters, the analysis started training and testing with the data from each subject. Figure 10 presents the boxplot distribution for the hit rates obtained for each subject from two perspectives: for overlap rate (a) and for the classifier (b).

The same behavior regarding the influence of the overlap rate could be observed in all subjects, as presented in Figure 10a. The distributions were concentrated close to the medians, with a low rate of outliers. Some subjects had a high increase in accuracy changing the overlap of the segments. This can be noted in subject 3, which exhibited an accuracy improvement from 40% to 90%. For other subjects, such as 6 and 7, the gain in accuracy was around 20%, with a hit rate of close to 100%. On the other hand, Figure 10b shows that some classifiers had more concentrated distributions, independent of the subject. This was the case with the RF and SVMRBF classifiers, which presented accuracies close to 100% for all subjects independent of the overlap of the segments. ELM and KNN demonstrated a large range in the distributions for more than one subject (i.e., subjects 3, 6, 9, and 12). In this way, these classifiers presented a higher dependence on the subjects compared to RF and SVMRBF.

This first analysis for the subjects took into account that the pattern recognition process uses only the individual data, as the training stage is a calibration process. Then, the second analysis for the subjects was performed, presenting the complete sequence of acquisition trials in the test step. The results of this analysis can be observed in Figure 11. It can be noted that this approach changed the behavior established as observed in the previous analysis, especially in Figure 11a for the data without overlap in segmentation. The SVMRBF, which was one of the classifiers with a high hit rate for a 100% overlap rate, presented accuracies ranging around 20%. Its hit rate only increased with the presence of overlap segments. At the same time, ELM, KNN, and RF classifiers maintained the same pattern observed for the previous analysis, meaning that they are less sensitive to changes in the trails. The Gaussians in SVMRBF could represent a high dependence for each subject, which explains their increase in the hit rates when more samples are present in the distribution by the overlap of the signal.

Using the best rate in overlap segments (12.5%), Figure 11b shows the obtained distributions for each subject for all classifiers. The ELM classifier presented high dispersion for all subjects, with accuracies ranging from 55% to 85%. The other classifiers presented high values with a concentrated distribution but with a high number of outliers. Some of them reached 20% less than the median in the subjects, especially SVMRBF. KNN and RF presented a similar behavior for all subjects, KNN having a low number of outliers and being close to 100%. However, their distributions did not present statistical difference (using a Wilcoxon rank sum test, with *p*-value > 0.05), and it established the execution time during the test step. The KNN classifier had an execution time to provide an output for each sample that was 200 times greater than the RF classifier. This was due to the nature of this classifier, also called a “lazy” classifier, because, for each sample, the Euclidean distance is calculated for all of the training data. For this reason, RF had some outliers at up to 87%; however, the accuracies had distributions above 98% for all subjects.

Using the obtained results for the RF classifier for the acquisition trails validation, Figure 12 presents the confusion matrix for the 26 classified gestures. The color bars show the accuracy in each cell, where green cells represent the high hit rates; red cells represent the misclassification samples; and the white cells represent where there was no relevant misclassification (less than 0.3%) values of correspondence. Moreover, the influence of the overlap rate, a parameter that demonstrated a high correlation with the accuracy, is shown. The disruptive windows presented a high dispersion on the hit rates, with an elevated number of false positives and false negatives, until in Figure 12c,d the classes that are more misclassified could be identified.

In general, the best-recognized gestures were “A”, “B”, “E”, “Q”, “W”, and “Y”, with high precision. Regarding gestures that were incorrectly classified, one can note that it occurred where the signals recruited similar movements. For example, this is the case for the gestures from the letters “F” and “T”, “C” and “O”, “P” and “Q”, and “Y” and “Z”, that were presented in Figure 1. The reason for this error was recruitment of similar muscle groups (occasioning similar sEMG signals in the channels). However, it presented accuracy values lower than 0.5%, indicating that the system obtained high hit rates for each sign.

Lastly, the data for men and women were separated to analyze if there was difference between them in the classification. In the average for the best classifier and feature set (G9 and RF) for the observed segmentation parameters (1.75 s of window length and 12.5% of overlap rate), men reached accuracies of around 0.9881 and women 0.9906. The Wilcoxon test (*p* < 0.05) was used to verify if the results allow the same distribution. For the first moment, the test did not find a significant difference between them, the *p*-value being higher (the *p*-value obtained was 0.461). However, due to the number of subjects, 10 men and two women, and close accuracy, it is not possible to conclude on a prevalence of performance between genders.

### 4.5. Comparison with Related Works

Aiming to compare the obtained results with those presented in the literature, Table 5 summarizes the main studies involving sEMG processing for SLRs that are correlated with the approach mentioned in this paper.

Before the discussion, it is necessary to briefly mention the obtained results. Regarding signal segmentation, window length and overlap rate parameters were evaluated. The overlapped segmentation was the parameter that most affected the accuracy of the classifiers; the disruptive window segmentation resulted in low accuracies. In the feature extraction stage, feature sets based on the literature were evaluated. In this work, a feature set (G9) formed by MFL, MNP, TTP, and RMS was proposed, which exhibited similar performance and reduced processing time for the literature feature sets. Concerning the classifiers, we began our investigation with nine classifiers and four of them exhibited performance above the average: ELM, KNN, RF, and SVMRBF. During the validation process in the analysis for each subject, RF demonstrated robustness with high accuracies and a short classification time for each sample. Overall, the pattern recognition process reached 99% for the RF classifier, with feature set G9, a window length of 1.75 s, and an overlap rate of 12.5%.

Comparing the number of signs, a closed set of gestures was considered, and comparing with the works that had Libras recognition as an objective for sEMG, 26 signs were used (more than the 20 used in [38,39]). Besides that, signals from 12 subjects (both male and female) were evaluated, higher than the similar works that explored between three and eight subjects. To contribute the findings regarding the individual classification to a generalist process, the validation stages were tested separately for all dataset, for each subject, and for the acquisition trials. The accuracies remained similar in all these processes.

Regarding the number of channels and acquisition mode, it is notable that the armband approach is a trend for gesture recognition, which has appeared also in SLR systems. Differently from fixed placement of the electrodes, the device is positioned with a single orientation, easing the process for people that are not specialist in anatomy or muscle localization on the forearm. Another proposition is the use of inertial unities (such as accelerometers) to aid the recognition process. In this work, only sEMG signals were considered, verifying this contribution in this process. It is right to affirm that inertial sensors can help this kind of application, but it is necessary to validate how much the sEMG supports Libras recognition.

In segmentation, a great impact on the hit rates was found, or in other words on the influence of the SLR. These parameters were not examined appropriately in the SLRs works. The works in Table 5 had a window length of between 200 and 750 ms, reaching accuracies between 80% and 98%. These values are lower than the values used in this work, where the best values were achieved for 1 and 2.25 s. We also searched for a generalist model (with data from more subjects), which was evaluated for more than one value of window length. Nevertheless, it is important to mention that for overlap rates of 25% and 12.5%, window lengths of 0.25 and 0.5 s reached accuracies close to 90% for the RF and KNN classifiers (as demonstrated in Figure 7). Moreover, overlapping signals may increase the number and variability of instances for the classification system training, which should be even more relevant for approaches that demand large datasets. Classifying in smaller steps with overlapping has a similar meaning of increasing the system resolution, as more data samples are processed in a given epoch of the time series. However, making the window step small may increase the processing time, and making the window size smaller would reduce the information within it. In this way, a trade-off should be reached between being feasible for real-time operation and having time resolution and classification accuracy.

Concerning the features, similar works applied some features and feature sets; however, they did not present their influence on the classification process. Both single features and feature sets approaches were evaluated in this work and the G9 set was considered suitable to be used for the armband approach in Libras recognition. Some evidence and hypotheses explain why the G9 set presented these values of the hit rate. RMS is closely related to MAV (features that are present in a large number of works in Table 3), which is a time-domain feature that provides amplitude information. MFL is a feature that has presented high accuracies for systems with higher sampling rates than the Myo^TM^ device [55]. TTP and MNP are frequency-domain features related to the energy of a power spectrum, and are complementary to the time-domain features. Moreover, these feature sets presented a lower computational cost compared to similar literature feature sets, being more suitable to be used in embedded systems.

A large number of different classifiers was evaluated in this work, in contrast to the works in Table 5 that applied around four classifiers. From the nine classifiers, four exhibited high accuracies for the alphabet in Libras: RF, KNN, SVMRBF, and ELM, in this order of accuracy. As in [27], the random forests exhibited the highest accuracy and was more suitable for an embedded application. Moreover, in this work, two methods of validation were tested: one based on the single subject test and another on an acquisition trial, the last being a generalist way to evaluate the signs.

## 5. Conclusions

This paper presented the recognition of Brazilian Sign Language and the influence of signal processing parameters during the classification process: signal segmentation, feature extraction, and pattern recognition. The sEMG signals were collected using an armband approach using the wearable device Myo^TM^. Data from 12 subjects were acquired and the 26 letters of the Brazilian Sign Language were classified. Only sEMG signals were used in this approach, verifying the contribution of this signal to this task.

The main purpose of this paper was to demonstrate how the data processing steps and parameters influence the SLR process, detailing the contribution of each stage to the accuracy. It was found that the segmentation was the parameter with more influence in the pattern recognition process, and especially the overlap rate. This can be explained due to the increase in the useful information and the instances for training, and it makes it suitable for embedded applications (if used together with the RF classifier). In overlapping, it is more important to consider for future developments time-dependent models and post-processing techniques, such as majority vote, to guarantee high accuracies.

The features were found to be a second factor of importance, mainly in the evaluation of the feature sets. It was demonstrated that features from both the time and frequency domains need to be used and it is possible to use a feature set with similar hit rates in the literature with low processing time, such as in the case of G9. Future works can explore the influence of the features on other segmentation parameters, detecting if there are significant changes.

In classification, RF was robust and presented high values in all the tests. Overall, a high accuracy (reaching 99%) was found. To test the robustness of this method, the separation of the test set for the acquisition trials was proposed and the system kept similar hit rates. The classes with more than 0.03% (indicated in Figure 12d) showed that the gestures with more confusion were the signs with similar movements, which represent similar muscle activations. Future works can explore the use of classifiers that are not dependent on handcraft feature selection, like CNN, deep learning techniques, and classifiers with time interval information, such as HMM.

The single sEMG armband approach for signal acquisition proved capable of recognizing signs from Libras. For future works, the database could be increased in terms of both number of gestures and signs, and the acquisition parameters could be analyzed, for example the number of channels; in addition, comparisons could be made with the fixed electrode approach and between men and women.

## Figures and Tables

**Figure 1 sensors-20-04359-f001:**
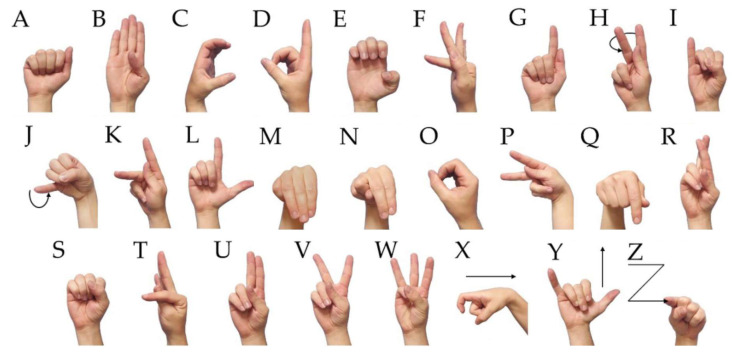
Alphabet set from Brazilian Sign Language (Libras). The set is composed of 26 letters, including 20 static gestures and six dynamic gestures.

**Figure 2 sensors-20-04359-f002:**
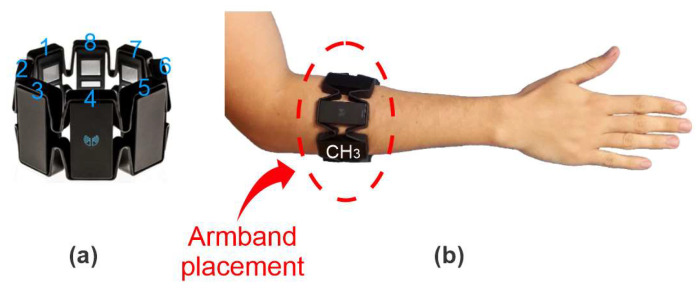
(**a**) Commercial Myo^TM^ device used for data acquisition with the numbers of its respective channels. (**b**) Placement of the armband on the subjects’ forearm. The reference (channel 3) was placed on the *flexor carpi ulnaris* to make all acquisitions uniform.

**Figure 3 sensors-20-04359-f003:**
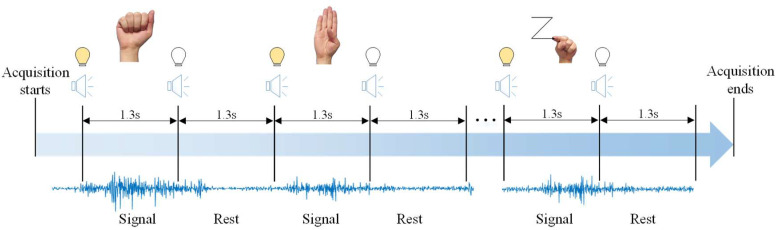
Sequence for sEMG signals acquisition. When the acquisition started, a light and a buzzer indicated that the subject should perform the gesture. After the 1.3 s, the light indicator turned off and the buzzer indicated to the subject to rest. After 1.3 s, both indicators turned on, repeating this process until the acquisition was completed.

**Figure 4 sensors-20-04359-f004:**
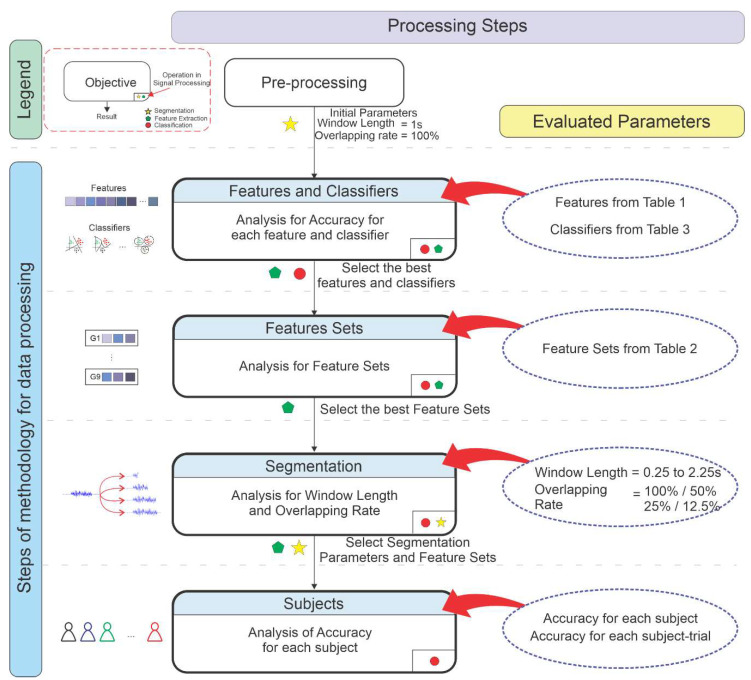
Steps in data processing methodology.

**Figure 5 sensors-20-04359-f005:**
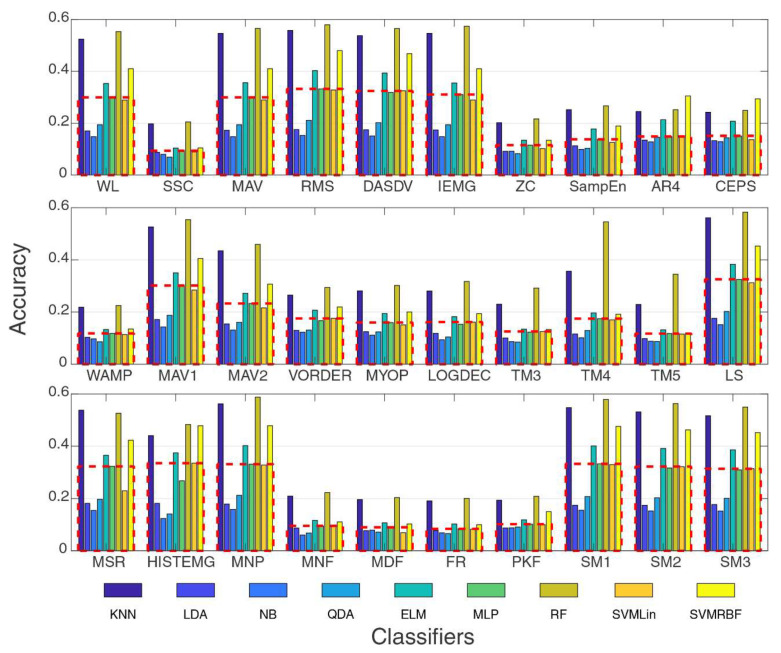
Accuracies for each extracted feature for all the classifiers considered in this work. Their abbreviations are listed in Table 1 and in Abbreviations section. As segmentation parameters, a window of 1 s and an overlap fraction of 1 (100%, a disruptive window without overlapping) were considered. The mean for each feature is presented by the dashed lines. The features with high accuracies were used for the creation of the feature sets G6 to G9.

**Figure 6 sensors-20-04359-f006:**
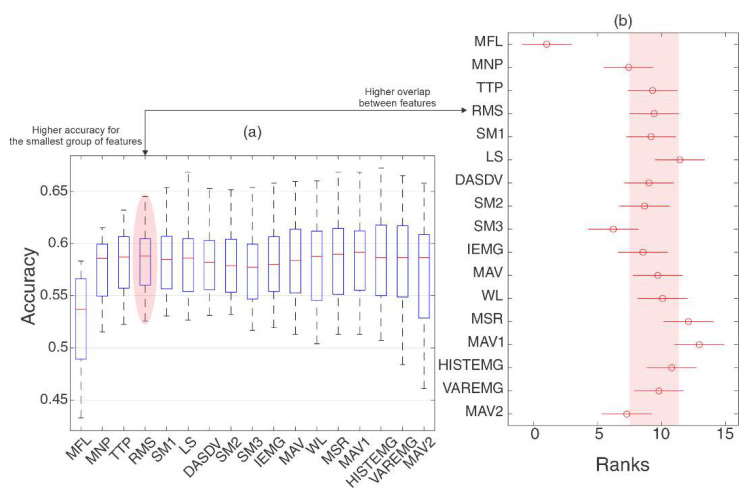
The selection process of the features from G6 to G9. The boxplot in (**a**) presents the distribution of accuracies increasing the number of features of G6 (from the highest to the smallest hit rate obtained separately) for the best classifiers (ELM, KNN, RF, and SVMRBF). The bars represent the 25th and 75th percentiles; the whiskers represent approximately 99.3% of the coverage of distribution, and the central mark represent the medians. (**b**) Distributions for the Tukey post-hoc from Friedman statistical test to evaluate the distribution of results with a confidence interval of 95% (*p*-value < 0.05) when increasing of the number of features in the classification process. The highlight indicates the distribution that is related to high accuracy obtained with a reduced number of features in (**a**) that corresponds statistically with the several distributions as demonstrated in (**b**).

**Figure 7 sensors-20-04359-f007:**
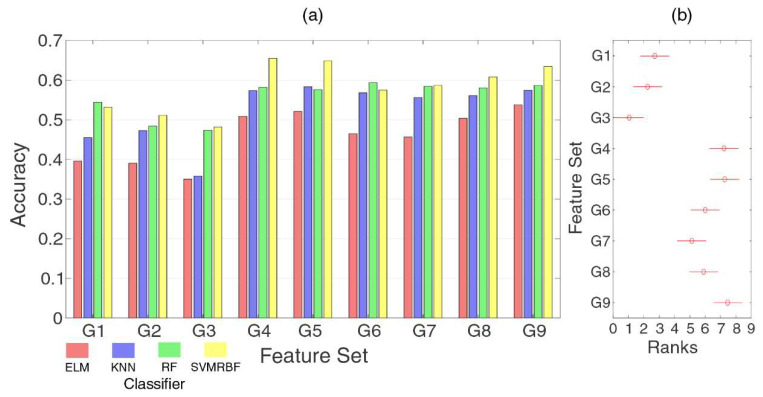
(**a**) Accuracies for the selected features sets (G1 to G9) for the classifiers that presented the highest accuracies in the previous analysis. (**b**) Distributions for the Tukey post-hoc from Friedman statistical test to evaluate the distribution of results with a confidence interval of 95% (*p*-value < 0.05). Two sets of ranks can be observed, separating G1 to G3 and G4 to G9. One can note that the feature sets G4, G5, G6, G8, and G9 have similar distributions.

**Figure 8 sensors-20-04359-f008:**
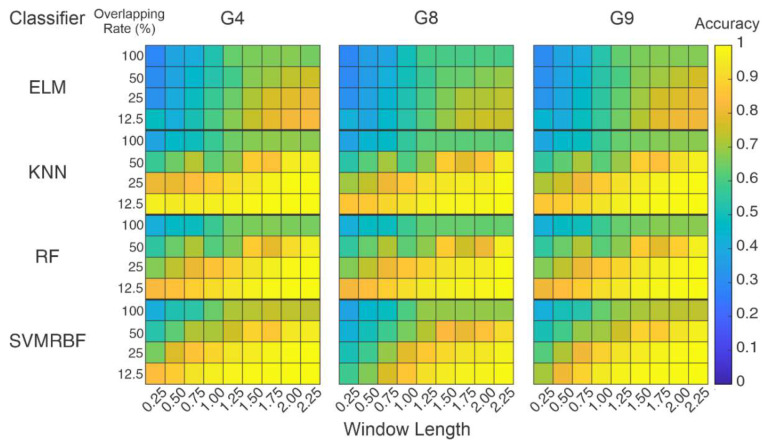
Results for the segmentation analysis. The color bar indicates the accuracy for each classifier on the influence of feature sets and segmentation parameters. The high accuracies are concentrated for large window length and small overlap rate.

**Figure 9 sensors-20-04359-f009:**
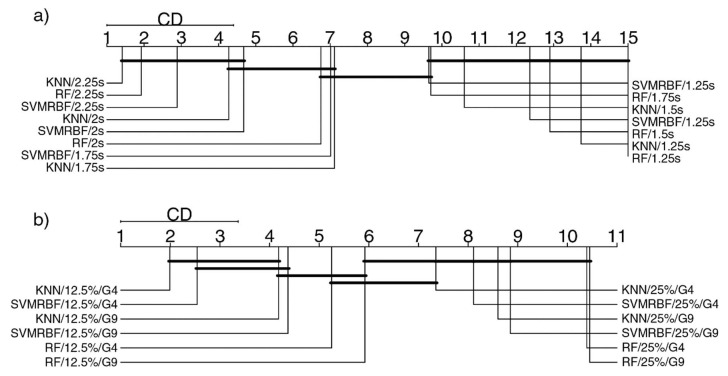
Critical distance diagram (CD) obtained from Friedman test and Nemenyi post-hoc test. The test was performed for (**a**) window length and (**b**) overlap rate variation. On the left, the best average rank performances are indicated. The critical difference attributes are denoted by the lines that connect the distributions. They indicate where there are no statistical differences. It can be noted that a window length of 1.75 s presented similar distributions to 2.25 and 2 s in several classifiers. Even the 12.5% overlap rate showed the best ranks, and some distributions were highly related to values of 25%.

**Figure 10 sensors-20-04359-f010:**
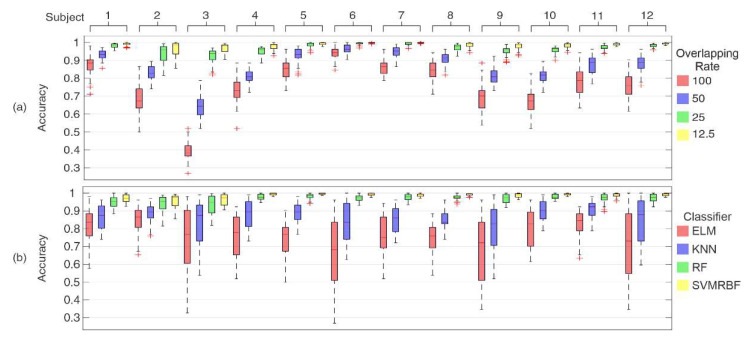
Distribution of accuracies considering the data for each subject individually in the test and training steps. (**a**) Distributions concerning overlap rate and (**b**) distributions concerning classifiers. The bars represent the 25th and 75th percentiles; the whiskers represent approximately 99.3% the coverage of distribution, and the crosses (+) are the outliers.

**Figure 11 sensors-20-04359-f011:**
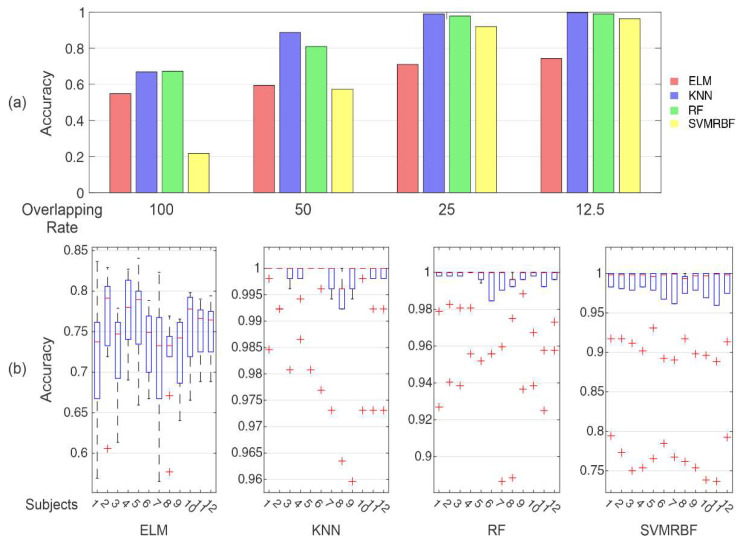
Hit rates obtained for acquisition trial validation. (**a**) Accuracies obtained for all subjects considering the variation of signal overlap related to classifiers. (**b**) Distribution of accuracies for an overlap rate of 12.5% (the best value observed in (a) for each subject and classifier). The bars represent the 25th and 75th percentiles; the whiskers represent approximately 99.3% of the coverage of distribution, and the crosses (+) are the outliers.

**Figure 12 sensors-20-04359-f012:**
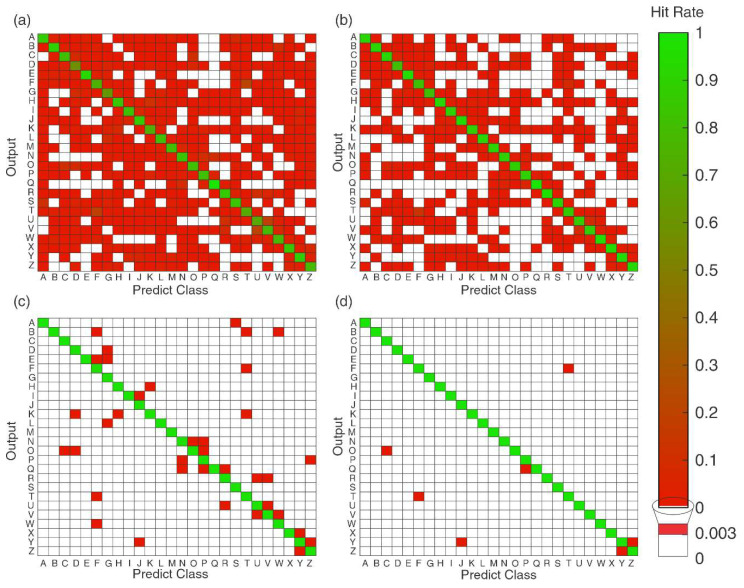
Confusion matrix for random forest classifier for the classes for all subjects using the validation by the acquisition trials, highlighting the influence of overlap rates of 100% (**a**), 50% (**b**), 25% (**c**), and 12.5% (**d**). The color bar represents the accuracy obtained in each case and the white cells represent values less than 0.3% of misclassification samples.

**Table 1 sensors-20-04359-t001:** Time-domain and frequency-domain features extracted from this work.

Domain	Feature	Feature Name	Parameters
Time	AR4	4th-order Autoregressive Coefficients	-
	CEPS	Cepstral Coefficients	4th-order
	DASDV	Difference Absolute Standard Deviation Value	-
	HIST	Histogram	9 Bins
	IEMG	Integral of EMG	-
	LOGDEC	Log Detector	-
	LS	L-Scale	Two Moment
	MAV	Mean Absolute Value	-
	MAV1 and MAV2	Modified Mean Absolute Value	-
	MFL	Maximum Fractal Length	-
	MSR	Mean Square Root	-
	MYOP	Myopulse percentage rate	Threshold = 10^−2^
	RMS	Root Mean Square	-
	SampEn	Sample Entropy	Dimension = 2 r = 0.2 σ
	SSC	Sign Slope Change	Slope threshold = 10^−4^
	TM3, TM4, and TM5	Absolute Value of 3rd, 4th and 5th Moments	-
	VAREMG	Variance	-
	VORDER	V-Order	3 Order
	WAMP	Willison Amplitude	Threshold = 10^−2^
	WL	Waveform Length	-
	ZC	Zero Crossing	Amplitude threshold = 10^−2^
Frequency	FR	Frequency Ratio	Low frequencies = 10–50 Hz High frequencies = 51–100 Hz
	MDF	Median Frequency	-
	MNF	Mean Frequency	-
	MNP	Mean Power Spectrum	-
	PKF	Peak Frequency	-
	SM1, SM2, and SM3	Spectral Momentum	-
	TTP	Total Power Spectrum	-

**Table 2 sensors-20-04359-t002:** Feature Sets organized and applied in this work.

Feature Set	Features
G1	Hudgins et al. (1993) set: MAV, WL, ZC, and SSC [54];
G2	Liu et al. (2007) set: AR4 and HIST [56]
G3	Most repeated features in Section 2 (Related Works): MAV and AR4;
G4	TD4 set: MFL, MSR, WAMP, and LS [55];
G5	TD9 set: LS, MFL, MSF, WAMP, ZC, RMS, IAV, DASDV, and VAREMG [55]
G6	Best features performed individually
G7	Best time-domain features in G6
G8	Best frequency-domain features in G6
G9	Reduced feature set from G6 with relevance in accuracy based on statistical analysis

**Table 3 sensors-20-04359-t003:** Classifiers applied in this work and their parameterization.

Classifier	Parameters
*k*-Nearest Neighbor (KNN)	1–nearest neighbor
Linear Discriminant Analysis (LDA)	-
Naïve Bayes (NB)	Normal distribution
Multi-Layer Perceptron (MLP)	30 neurons in hidden layer
Quadratic Discriminant Analysis (QDA)	-
Random Forest (RF)	30 trees
Extreme Learning Machine (ELM)	1000 neurons in the hidden layer
Support Vector Machine with Linear Discrimination (SVMLin)	C = 100
Support Vector Machine with Radial Basis Function Discrimination (SVMRBF)	C = 10 and Gaussian size = 1

**Table 4 sensors-20-04359-t004:** Accuracies from the classifiers with similar accuracies in segmentation analysis (KNN, RF, and SVMRBF). The range of 1.25 to 2.25 s for window length and 25 and 12.5 overlap rates were considered due to the similar results reached in the segmentation influence analysis.

	Accuracy (%)					
Window Length	G4					
	KNN Overlap Rate		RF Overlap Rate		SVMRBF Overlap Rate	
	25	12.5	25	12.5	25	12.5
1.25 s	92.5 ^ab^	98 ^ab^	89.3 ^ab^	94.9 ^ab^	94.2 ^ab^	98.3 ^abc^
1.5 s	95.1 ^ab^	99 ^abc^	93.8 ^abc^	97.5 ^abc^	96.4 ^abc^	99.1 ^abcd^
1.75 s	97.8 ^abc^	99.7 ^bcd^	97.2 ^cd^	99 ^bcd^	98 ^cd^	99.5 ^abcd^
2 s	99.8 ^abceF^	99.9 ^cdeF^	98.9 ^cde^	99.6 ^de^	99.3 ^cde^	99.8 ^bcde^
2.25 s	99.9 ^ceF^	99.9 ^cdeF^	99.9 ^deF^	99.9 ^deF^	99.8 ^de^	99.9 ^de^
	**G9**					
1.25 s	93.2 ^ab^	97 ^ab^	90.3 ^ab^	95.3 ^ab^	93.8 ^ab^	97.7 ^ab^
1.5 s	95.8 ^abc^	98.8 ^abcd^	94.3 ^abc^	97.7 ^abcd^	96.3 ^abc^	99 ^abc^
1.75 s	97.5 ^bdc^	99.4 ^bcde^	96.9 ^bcd^	98.7 ^bcde^	97.9 ^bcd^	99.4 ^bcd^
2 s	98.8 ^cde^	99.5 ^bcde^	99.2 ^cde^	98.2 ^bcde^	98.9 ^cde^	99.6 ^cde^
2.25 s	99.9 ^deF^	99.9 ^cdeF^	99.9 ^deF^	99.9 ^cdeF^	99.7 ^de^	99.9 ^de^

^a,b,c,d,e^ Equivalent statistical distribution for each window length in its respective column (same overlap rate) from Tukey post-hoc from Friedman test (*p* > 0.05): 1.25 (a), 1.5 (b), 1.75 (c), 2 (d), and 2.25 (e) seconds. ^F^ Equivalent statistical distribution for each 25 and 12.5 overlap rate pair in its respective row according to Wilcoxon test (*p* > 0.05).

**Table 5 sensors-20-04359-t005:** Sign Language recognition (SLR) systems with sEMG as input and their main characteristics.

Work	SLR	Signals/ Subjects	sEMG Channels and/Other Sensors	Window Length/ Overlapping	Features	Classifier	Results
[25]	ASL	9/-	2	512 samples/Y	IAV, ZC, DAMV, AR, CEPS, Mean frequency	DA	97%
[10]	GSL	60/3	5/Accelerometer	-/-	Sample Entropy	DA	93%
[24]	CSL	72/2	5/Accelerometer	100 ms/N	MAV, AR	DT, HMM	98%
[29]	CSL	223/5	4/Accelerometer	64 points/Y	MAV, AR	DTW, HMM	92% to 96%
[28]	ASL	40/4	4/Accelerometer Gyroscope Magnetometer	128 ms/N	MAV, AR, HIST, RMS, Reflection Coefficients, VAREMG, WAMP, MDF, Modify MNF	NB, KNN, DT, SVM	96%
[27]	CSL	121/5	4/Accelerometer	128 ms/Y	MAV, AR, Mean, VAREMG, Linear Prediction Coefficients	RF, HMM	98%
[11]	ASL	80/4	4/ Accelerometer	128 ms/N	MAV, AR, HIST, RMS, Reflection Coefficients, VAREMG, WAMP, MDF, Modify MNF	NB, KNN, DT, SVM	96%
[38]	Libras	20/-	8 (Myo armband)	-/-	Mean	SVM	41%
[26]	KSL	30/6	8 (Myo armband)/ Accelerometer Gyroscope Magnetometer	200 ms/N	Raw signal	CNN	98%
[23]	CSL	18/8	5/ Accelerometer	176 ms/Y 50-350 ms/Y	MAV, ZC, SSC, and WL	LDA	91%
[39]	Libras	20/1	8 (Myo armband)	750 ms/N	IEMG, MAV, RMS, LOGDEC, ZC, SSC, MNF, PKF, MNP, SM0, SM1	MLP	81%
This Work	Libras	26/12	8 (Myo armband)	Window Length: 0.25 to 2.25 s/ Without overlap, 50%, 25%, and 12.5%	AR4, CEPS, DASDV, HIST, IEMG, LOGDEC, LS, MAV, MAV1, MAV2, MFL, MSR, MYOP, RMS, SampEn, SSC, TM3, TM4, TM5, VAREMG, VORDER, WAMP, WL, ZC, FR, MDF, MNF, PKF, SM1, SM2, SM3, TTP	KNN, NB, LDA, QDA, RF, ELM, MLP, SVMLin, SVMRBF	99%

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
