# Peer review of "Analysis of Influence of Segmentation, Features, and Classification in sEMG Processing: A Case Study of Recognition of Brazilian Sign Language Alphabet"

_sensors, 2020, doi:10.3390/s20164359_

Round 1

Reviewer 1 Report

The subject of the paper is to carry out an exhaustive analysis of the influence of the segmentation, feature extraction, and classification in the recognition of sign language.

The authors used the Myo armband device to the data acquisition phase.  Authors perform an exhaustive and pertinent analysis on a data set for sign language identification. Even though the study was carried out on a very limited set of signs and with most of the static signs, this paper can help to understand the influence of the parameters on the recognition or identification of signs.

The idea of the proposed method is very interesting and the experimental results sound feasible. The paper's written presentation is good, and easy to follow. In general, the scientific quality of the paper and its relevance in the field is good.

Author Response

We would like to thank you for your review.

Reviewer 2 Report

The paper is well written and easy to read. 

The only key comment I have would be for the author to discuss about how = deep learning approaches would work with their signal processing. I understand that the small amount of initial data from only 12 subjects would make it difficult to train a deep learning based model at the moment, but most state of the art in analog signal to digital transformation are done with a variant of recurrent neural net architecture. Like deep speech in speech recognition. 

I would also ask the author to provide the code and data on Github if there is no privacy concern. 

I personally believe that this publication would be of interest to most readers in terms of the impact of the application. 

Author Response

We would like to thank you for yours review. We inserted in the future works about the use of deep learning for future works. First, we need to increase our database and use techniques of data augmentation to support this future analysis. About the database, we appreciate the suggestion. We will be studying the best way to publish the data.

Reviewer 3 Report

The article presents a method for the recognition of Brazilian Sign Language gestures using surface electromyography (sEMG) signals acquired from the armband device. The paper is interesting and well-prepared. However, the following items should be addressed properly in the revised version of the manuscript, before it could be considered for acceptance and publication.

Comments:

  1. All abbreviations used in the abstract should be explained.
  2. The introduction focuses on the use of sEMG signals for gesture recognition. The challenges and limitations of using sEMG should be discussed, supported by proper literature sources.
  3. The novelty and contribution of the paper over existing works should be explicitly stated at the end of Section 1.
  4. In Related works Section 2, a discussion on the most recent papers in this area, such as Recognition of american sign language gestures in a virtual reality using leap motion, and Hand sign language recognition using multi-view hand skeleton, is required. The section must be concluded be a summary of related works presented in a tabular form and enlisting the methods, signal features, and classification methods used, and the results (accuracy) achieved.
  5. Discuss the placement of the device in more detail. Was it placed on a left hand or a right hand? How many subjects were right-handed and left-handed?
  6. For the histogram feature, provide motivation for using 9 bins. For autoregression features, motivate the use of 4th order AR. For cepstral coefficients, provide a number of coefficients used. Explain less common features (such as MAV1 and MAV2, V-order) by presenting a mathematical expression of calculation and giving a reference to papers, which introduced them.
  7. The cross-validation procedure should be explained in more detail. Did you perform cross-subject classification or intra-subject classification? Were the data from the same subjects present both in training and testing data?
  8. Correlation analysis of features should be performed. Which features are highly correlated and how it influences the formation of feature sets?
  9. Statistical analysis in section 4.3: also perform the post-hoc Nemenyi test and present the mean ranks of method/window/featureset combinations using a critical distance diagram.
  10. Are there any statistically significant differences between genders in the results?

Author Response

Dear reviewer,
We are very grateful for your considerations in this work. Each topic is discussed in attached file. We merge in the review letter the highlighted version.

Round 2

Reviewer 3 Report

The authors did an excellent job in improving the paper according to my comments. I have no further comments and recommend the paper to be accepted.